# Radical Composition and Radical Reaction Kinetics in the Probe-Irradiated XLPE Samples as a Potential Source of Information on Their Aging Degree

**DOI:** 10.3390/ma15165723

**Published:** 2022-08-19

**Authors:** Hanna Lewandowska, Jarosław Sadło

**Affiliations:** Centre for Radiation Research and Technology, Institute of Nuclear Chemistry and Technology, 16 Dorodna St., 03-195 Warsaw, Poland

**Keywords:** cable insulation, aging, ionizing radiation, electron paramagnetic resonance, free radicals, oxidative degradation, polymers

## Abstract

Polyethylene is a model polyolefin, and a widely used material for the manufacture of many products, including cable sheaths. Understanding degradation mechanisms at the atomic scale leading to oxidation during aging is crucial for many long-term applications. The concentrations of radicals derived from oxidation and chain scission during radio-oxidation, as well as their ratio, are important parameters controlling the predominance of chain scission or crosslinking of the polymer. In this work, we propose a cryogenic EPR technique for measuring oxidation- and fragmentation-derived radicals as a less-destructive method for the evaluation of cable insulation aging and performance capability. We investigate the effect of the low-dose and high-dose radiation aging on the formation of free radicals in the polymer matrix that are both unprotected and protected by antioxidants. The stability of radicals after aging is a determinant of macroscopic processes and structural changes during aging. Under the conditions of the higher dose rate, the peroxy radical buildup is lower per dose. Peroxy radical buildup is followed by decay during aging, in accordance with POOH content. Our results allow the prediction of the capability of the antioxidant to protect the XLPE material in the function of dose and time.

## 1. Introduction

The ubiquity of free radicals in polymer systems has made EPR spectroscopy an essential tool in polymer science, alongside other major analytical techniques. The scope of application of EPR spectroscopy has expanded, especially due to the growing trend to produce polymers with enhanced stability, follow the charge transfer process in conducting polymers, learn more easily about reaction mechanisms and kinetics of complex polymer reactions, and to perform comprehensive structural and conformational analyses. EPR spectroscopy is used in polymer science in a variety of ways, including, but not limited to, studying the structure, conformation and dynamics of polymer chain segments, investigating degradation, defects, charge transfer properties, free-radical reaction kinetics and mechanisms, and EPR imaging of the polymer matrix [1].

The importance of nuclear power has risen again in the face of the energy crisis and the need to diversify energy sources [2]. A reversal of the trend of nuclear phase-out in power generation, which is a serious alternative to fossil fuels, can be expected. Therefore, it is all the more important to be able to assess the condition of nuclear reactors. 

In the development of nuclear power, a particularly strong emphasis is continually placed on developing newer and newer non-invasive methods to ensure the safe operation of nuclear reactors [3]. 

The current design life of NPPs could potentially be extended to 80 years [4]. During this extended lifetime, all safety and operationally relevant instrumentation and control systems must meet the criteria for safe and reliable operation, both during normal nuclear plant operation and during and after accidents. This in turn requires an appropriate qualification and aging control program for cables with special nuclear power applications [3].

On average, about 25,000 cables with a total length of 1500 km are installed in a nuclear power plant [5]. It is known that instrumentation and control (I&C) system cables with safety functions and cables that provide power plant availability can degrade over their lifetime due to aging from environmental factors such as temperature, radiation, moisture, and vibration. Maintaining the ability of I&C cables to perform their intended functions, therefore, requires various minimally invasive techniques to monitor their condition. These techniques can potentially be used to estimate the remaining service life based on the relationship between condition indicators and aging stressors, thereby supporting a preventive and effective aging management program. 

Recommendations for a management system for the qualification and monitoring of cable ageing in NPPs have been published by the IAEA [3]. Currently, the gold standard for determining cable insulation degradation is to measure elongation at break (EaB) with the assumption that the EaB must be higher than 50% for the aged material to pass the test [6]. However, this ex-situ measurement requires the removal of a sample of considerable size for laboratory testing. To objectively determine the serviceability of cables, reliable non-destructive or low-volume examination methods are desirable. Many tests are available to evaluate various aspects of the electrical and mechanical performance of cables, but no single test is suitable for all cables, nor does any single test conclusively confirm the condition of a cable’s polymeric material. Currently developed cable condition monitoring tools assume a multi-scale evaluation based on tests obtained by different methods [7]. Therefore, it is important to develop new methods for cable condition parameterization in order to develop multi-scale aging models. Methods that give insight into the actual chemical processes occurring in the polymer allow us to reach a much deeper understanding of the observed macroscopic effects [6].

Free radicals are the basis of processes occurring during the aging of polymers, and determine the resulting behavior and mechanical and electrical properties of insulations. Therefore in this work, we propose a cryogenic EPR technique for measuring oxidation- and fragmentation-derived radicals, in particular superoxide (POO•) and alkyl (Alk•) radicals as a new method for evaluation of cable insulation condition. The degradation of a silane cross-linked polyethylene (XLPE) matrix, protected or unprotected with antioxidants, was performed in the framework of the H2020-Euratom TeamCables coordinated project. These model insulation materials were subjected to radiochemical ageing at two different dose rates (8.5 and 400 Gy·h^−1^) in air at low temperatures close to ambient (47 and 21 °C, respectively). We analyzed the occurrence and interconversions of free radicals in the samples obtained after each aging period and related the observed radical processes to the consequent physicochemical changes. A special advantage of EPR application in material testing, in accordance with the mentioned requirements for non-destructive testing, is that this method requires very small sample sizes in the order of tens of mg. This eliminates the need to obtain large fragments of cable.

### Research Design and Approach

An important branch of radical reactions in polymers are the reactions connected with the process of oxidation. These interactions with oxygen result in an increased concentration of polymer alkoxy and peroxy radicals. Fragmentation reactions of oxygen-centered radicals yield new oxidation products with structures not found under an inert atmosphere [8]. These radicals can proceed to undergo abstraction, fragmentation and recombination reactions both with the original polymer and other products of decomposition. Such reactions can affect the polymer during processing, particularly if the temperature required is high, and also its performance during its end-use [9]. To understand these reactions, the so-called Bolland and Gee reaction scheme with its subsequent developments has been applied to explain the chain reaction characteristics of both thermal and radiation-induced oxidation of polyolefins [10]. Antioxidants can be added to the plastic formulation to inhibit the mentioned effects. Antioxidants function by interfering with the radical oxidation and degradation reactions [11].

The need for antioxidants in polymeric materials has been known for many decades. They inhibit the oxidation of other molecules by reducing intermediate free radicals and preventing the accumulation of reactive oxygen species. Therefore, the presence of antioxidants is undeniably essential in any product. However, the question remains open: what are the long-term effects of antioxidants? Are they able to give products the necessary long-term protection to prevent degradation during long-term use in strategically important facilities such as nuclear power plants? EPR is a tool for assessing antioxidant activity [12]. It allows the effectiveness of antioxidants to be determined by taking into account their chemical activity and kinetics, and the effect of a typical stress test on their performance.

Free radicals occur often in many reactions in polymers, but it has to be underlined that at room temperature most of the free radicals recombine or take the lowest possible energetic states, making the observation and analysis of the radical processes difficult. Even though radical processes constantly occur in the polymer, radicals are transient intermediate products and their actual concentrations at a given time are very low [13]. Therefore, in order to observe these radical-dependent reactions, a special technique has to be implemented. 

Electron paramagnetic resonance (EPR) spectroscopy, is a unique and very selective method dedicated to the observation of only paramagnetic species in the sample. The EPR-based method for investigation of the polymer structure, used in the herein-presented paper consists in combining low-temperature probe irradiation with the low-temperature EPR method. Pre-freezing of the sample (below 100 K) reduces the molecule motions and thus allows the slowing down of the interconversion kinetics for free radicals that will be produced by radiation in matter [14]. In this regard, gamma radiation is the most convenient type of ionising radiation, as it penetrates the matter uniformly in high ranges, inducing the uniform effect in the studied sample stored in a Dewar flask containing liquid nitrogen. After irradiation, the EPR spectra are recorded stepwise, at gradually elevated temperatures from 100 K to 310 K. This way it is possible to observe the processes of radical recombination, along with the gradually increasing kinetics of these processes, so both very fast processes and those engaging more stable individuals can be analyzed. Ionising radiation can be an aging factor for polymers in various applications, but in this particular approach, it is also applied as a probe for the rapid determination of aging processes in polymers.

Despite the relatively simple elemental composition, polyethylene is a fairly complex polymer due to its various structural and morphological forms. The presence of antioxidants or other additives significantly changes the radiation/thermal induced oxidative degradation. In the project, polyethylene was stabilised against thermo-oxidative or radiation-induced degradation over its entire service life under severe conditions with a primary antioxidant (AO), Irganox 1076 and a secondary stabiliser, Irganox PS802, which are capable of scavenging free radicals or decomposing hydroperoxides that contribute to aging processes. The mechanism of antioxidant reactions is not fully understood. In our previous work, the interpretation of EPR signals allowed us to determine the paramagnetic species participating in these reactions [15]. 

As previous studies have shown, radical processes differ significantly at the beginning and the end of long-term radiation aging, and are also dependent on other factors such as temperature, the type of aging factor, and above all the availability of oxygen during aging [8,16]. The oxygen availability in the bulk of the aged material is strongly dependent on both the temperature and the intensity of the aging medium and related to oxygen diffusion rate [17]. Chemical changes occurring in the material during the aging process cause the disappearance of the original reactive sites susceptible to radical reactions and the appearance of new ones. Thus, ionising radiation will cause the formation of different free radical signals in the aged and unaged material. Therefore, it can be assumed that the radical composition of the probe-irradiated materials will relate to the aging degree and that the observed reactions will show altered kinetics. Such a careful analysis of the EPR signals is a potential source of information on the aging degree of the tested polymer material. An objective of the presented study was to verify this hypothesis by comparing the probe-irradiation-induced radical processes observed in polyethylene after stepwise radiation aging for low and high dose rates. These processes were compared for both AO-protected and non-protected XLPE samples.

## 2. Materials and Methods

Two model materials were used in this study: the silane-crosslinked polyethylene (XLPE) and the silane-crosslinked polyethylene protected by antioxidants (AO-XLPE). They were manufactured as thin tapes (0.5 mm thick) as described in [18]. Two types of antioxidants (AO) were used in the AO-protected material: 1phr Irganox 1076, inhibiting the propagation of free radicals which are stabilised by its aromatic ring and 1phr Irganox PS 802, inhibiting the formation of hydroperoxides by thiol-mediated one-electron oxidation reaction. The impact of these additives on the aging behavior of XLPE has been studied in two radio-oxidant conditions close to room temperature (low and high dose rates). The corresponding accelerated aging treatments took place in ÚJV (Řež, Czech Republic). The tapes were placed in a metallic cylinder in the irradiator (Panoza for low dose rate or Roza for high dose rate) and maintained at low temperatures close to ambient, that is 47 °C (320.15 K) for the low-dose aging and 21 °C (294.15 K) for the high-dose aging. The samples were turned over every three weeks to ensure the homogeneity of the integrated dose in the whole sample and withdrawals took place at constant time intervals (Table 1). 

**EPR measurements.** The unaged and aged samples were placed in quartz tubes, frozen in liquid nitrogen and probe-irradiated with gamma rays in a Gamma Chamber 5000 (BRIT, India) up to a dose of 10 kGy at a dose rate of 2 kGy/h. Tests were conducted using an EMXplus Bruker (Bruker, Rheinstetten, Germany), a cw EPR X-band spectrometer equipped with an ER 4131VT temperature control system (Bruker, Rheinstetten Germany), with the following parameters: sweep width 35.0 mT, microwave power 10.0 mW, 1.0 mW or 0.1 mW, time constant 1.28 ms, 10 ms conversion time and modulation amplitude 0.1 mT. Quartz tubes filled with samples were placed in a cylindrical resonant cavity with high sensitivity. The spectra were measured every 30 K, from 100 K to the temperature at which the complete decay of radicals took place, using a controlled temperature annealing procedure. After heating, the samples were kept at the selected temperature for 5 min to achieve thermal equilibrium. The spectra were analysed using WinEpr software (v. 2.22 rev 12, Bruker Analytik, Rheinstetten Germany). The relative peroxy and alkyl radical concentration was determined by comparing the height of their characteristic spectral components (see below). 

The full record of spectra for tapes made of XLPE and XLPE protected with both antioxidants (AO-XLPE) in the subsequent aging periods after probe-irradiation with 10 kGy dose of gamma radiation and subsequent cryogenic EPR measurement are given in Appendix A.

The data presented here constitute part of a database of European Tools and Methodologies for an efficient aging management of nuclear power plant Cables project.

## 3. Results

The EPR spectrum of the studied non-protected XLPE sample subjected to a test dose of gamma radiation (Figure 1) allows several paramagnetic species to be revealed. The most abundant intermediate is a secondary alkyl radical, characterised by a sextet of hyperfine splitting (hfs) of about 3.0 mT. The second radical, demonstrating a septet of hyperfine splitting (HFS) of 1.4 mT with further complex splitting, probably in the form of a doublet, was assigned to allyl type radical. This signal can be recognised only for the unaged XLPE samples, both probe-irradiated in air and vacuum, up to the temperature of 220 K. Carbon-centered radicals were converted to a peroxy-radical, characterised by the center g-value of 2.014 and a characteristic asymmetric signal with spectral components at g-factors 2.0318 and 2.0095. Peroxy-radicals are able to abstract hydrogen from methylene groups of the main chain forming hydroperoxides that are converted to carbonyl or carboxylic groups and initiate chain oxidative degradation.

At 100 K for XLPE degraded by radiation (second interval i.e., dose 51 kGy, low dose rate), the dominant paramagnetic product is peroxy radical POO•, whereas the contribution of alkyl radical Alk• is much smaller than for unaged material. After thermal annealing of the samples, alkoxy radical –HCO• can be observed, being a secondary product of alkyl radical chain reactions (g⊥ = 2.0210) [13]. The signal appears at around 280 K in the spectra registered at 10 mW (Figure 2). 

Comparison of Figure 1A and Figure 2A (XLPE before and after aging) allows us to observe that the radical signals are more stable with temperature after aging. For the unaged sample, signals disappear above 280 K, while they are still intense at 310 K in the aged samples. According to Hettal et al. [19] the crystallinity of the same samples increases during aging, approximately to the same degree per dose of radiation in both aging regimes for the non-protected XLPE. The crystallinity increase effect may explain the change in stability of radicals, both oxygen- and carbon-centred, as it is known from the literature that radicals in the crystalline phase are significantly more stable as a result of inhibition of chain motions [15]. 

Thus, the signal of the irradiated XLPE is a superposition of several signals, deconvolution of which is difficult, but the intensity of the characteristic signal components that are least obscured by other signals can be used as an indicator of the content of individual radicals. In previous studies, it was determined that the most favourable conditions for the observation of carbon-centered radicals are in low microwave (MW) powers (0.1 mT), while peroxy radical (POO•) signals are pronounced at higher (10 mW) MW powers, due to partial saturation of carbon-centered signals [13]. For the sake of signal comparison, the conditions allowing the best differentiation between the signal coming from peroxy radicals and alkyl radicals, are at 0.1 mW microwave power with other parameters as described above. The choice of the characteristic intensities was made for POO• and Alk• and is justified below.

Figure 3 shows the comparison of low-power (0.1 mW, A) and high-power (10 mW, B) spectra of the non-treated, unaged polyethylene, probe-irradiated in vacuum (vac) and an air atmosphere (air), superimposed with the spectra of the aged samples after the second period of low-rate radiation aging, registered in 100 K. The subtraction of the spectra of the non-treated and aged samples at 100 K reveals the characteristic spectrum of the peroxy radical. Its component at g = 2.0318 (327.54 mT in the measurement conditions) to the lowest degree overlaps with other signals, and can be an indicator of POO• concentration. It is impossible to completely deconvolute the POO• signal. The most pronounced POO• component comprises also the Alk• component. It will be further referred to as (POO• + Alk•). The alkyl radical signal component at g = 1.9692 (337.9 mT in the measurement conditions) is the least influenced by the signals from other paramagnetic centers and will be further treated as an indicator of Alk• presence (see Figure 3). Some traces of peroxy radical can be already found in untreated samples, regardless of the irradiation atmosphere, i.e., in vacuum or in air (Figure 3).

Observation of changes in the overall signal intensity and the absolute and relative changes in the intensity of individual radicals as a function of temperature for different aging points allows for the tracking of the interconversions between oxygen-centered and carbon-centered free radicals. For unprotected, unaged XLPE, the overall signal is initially stable (for 100 K and 130 K). Above this temperature, for 160 K, a significant increase in signal is observed, presumably at the beginning of the glass transition [20]. At 250 K a dramatic decrease in the overall signal can be observed and the decrease in the component at g = 2.0318 (327.54 mT in measurement conditions) can be seen. These pronounced changes at 160 K and 250 K can be mainly linked to the formation and decomposition of the POO• radicals, respectively (Figure 3A). The alkyl radical signal in turn, after a slight increase between 110 K and 130 K, remains constant up to 220 K, after which it decreases, along with POO• decomposition. Relatively, an increase in the concentration of POO• relative to the alkyl radical is observed up to 250 K. Above this temperature, the concentrations of both radicals are minimal. The relative proportion of peroxy and alkyl radicals changes with the increase of the temperature of spectra measurement, revealing the occurrence of the buildup and decay processes resulting from the location of the radical center in the energetically favourable states and further translation and recombination of uncoupled electron-bearing sites (Figure 4A).

For the unaged non-protected XLPE samples probe-irradiated in vacuo, it can be seen that the decay of both components, at g = 1.9692 and g = 2.0318, has the same pattern, indicating that the decay of virtually only the Alk• radical is observed (Figure 4B). This decay as the temperature is raised for the sample irradiated in vacuum is monotonic from 100 K. Comparison of the Alk• radical decay pattern in vacuum with that for the sample irradiated in air, for which the formation and decay of POO• and the stabilisation of Alk• levels are seen, indicates that the stabilisation of Alk• levels at temperatures up to 250 K is a composite of Alk• recombination effects and the formation of these radicals as decay products of POO•, the peroxy radicals being the precursors of new secondary alkyl radicals. Alk• can be formed as a result of hydrogen abstraction and also by recombination of other radicals present, including POO•. 

For the second phase of aging of material unprotected with AO (52 kGy, low-rate, Figure 5), the highest concentration of POO• radicals with respect to Alk• among all the studied aging points is observed. While in unaged XLPE, the peroxy radical forms at a certain temperature, in aged XLPE it occurs from the lowest temperatures. Even though free radicals are present in the unirradiated samples only in trace amounts [13], after sample irradiation of the aged samples (both regimes) a maximum level of radicals, both peroxy and alkyl, is observed immediately at 100 K, which then decreases with annealing. It follows that in the aging process the corresponding paramagnetic centers are formed, which immediately after irradiation are transformed into POO• radicals. This pattern differs from the rise and fall of radical concentrations characteristic of unaged samples. In turn, the signal of allyl-type radical, seen for the unaged polyethylene in air and vacuum, is unnoticeable for the aged samples. These data suggest that radiation-induced free radicals form upon probe irradiation in oxygenated moieties already present in an aged sample. These may include the POOH groups, whose concentrations have been similarly characterised in samples from the same series by Colin et al. [19], who also observed that POOH concentrations increase in samples non-protected by AO, reaching their highest concentrations precisely during periods 2 and 3 of low-dose aging.

For the AO-protected samples, the formation of POO• is highly reduced in favor of the Alk• radical intensity, dominating the spectrum (spectra included in Appendix A). Alk• radical in the untreated sample is consumed with increasing temperature, which points to its recombination with AO (Figure 6). The peroxy radicals are extensively produced above 160 K, as shown by the inspection of the spectra for unprotected XLPE. Thus, the POO• concentration in AO-XLPE is a superposition of its recombination with AO and production; their concentration tends to drop above 180 K.

The relative concentration of (POO• + Alk•) versus Alk• is a measure of the relative content of peroxy radicals versus alkyl radicals. It gives information on the presence of reactive oxygen adducts in the polymer matrix at a given aging stage, which can be further reorganised to form alkyl radicals, causing further branching reactions and massive chain scission processes. For further analysis, we chose a temperature of 160 K, for which the maximum concentrations of the POO• radical are observed before it undergoes further recombination to end products (see Figure 1, Figure 2, Figure 4, Figure 5 and Figure 6). Analysis of the relative intensity of the (POO• + Alk•) to Alk• signal ratio by measuring the intensity at g = 2.0318 and g = 1.9692, respectively, can provide a measure of oxidative damage occurring with aging and allows us to determine the protective effect of AO. As can be seen in Figure 7, the relative proportion (POO• + Alk•)/Alk• changes with the degree of aging of the material. It can be observed that the buildup and decay differ depending both on the aging radiation dose rate and the presence of antioxidants. 

In Figure 7, large increases in POO• intensity are visible for the first and second aging periods for the materials non-protected by antioxidants. For the low dose rate, the initial accumulation of POO• vs. Alk• is followed by its decrease in longer aging times. This indicates that sites for POO• formation are produced during aging and then gradually consumed, probably due to further oxidation damage to these sites, possibly leading to the formation of terminal non-paramagnetic oxidation products such as R=O. Such a decrease in POO• at aging is not visible for samples aged at a high rate. One can assume these sites were consumed to a much lower degree in these specimens as compared to the low-radiation aging rate, because during aging at a higher rate, the consumption of these sites (into terminal non-paramagnetic oxidation products) along with the delivered dose was limited by the oxygen diffusion. Thus, for high dose rate, a gradual increase in POO• with respect to Alk• along with the dose can be seen.

For the AOXLPE, a pronounced protective effect of the antioxidants can be noticed. The POX intensity (in relation to Alk•) is much lower than in the case of unprotected specimens. The presence of AO significantly reduces the level of POO• to Alk• radicals. However, once the 202 kGy dose is reached, there is an increase in this ratio to values characteristic of non-protected XLPE. This illustrates the effect of depletion of the protective properties of AO for doses above 202 kGy.

It is also worth noting the (POO• + Alk•/Alk•) ratio curve as a function of dose for the low-dose regime (Figure 7B). Interestingly, the level of formed peroxy radicals is relatively much higher per dose than for the high-dose regime. In this case, the longer time at which aging takes place and, consequently, the better oxygen saturation in the diffusion process and the contribution of thermal aging during incubation over a longer period should be taken into account.

## 4. Discussion

To test a material’s susceptibility to aging factors, accelerated aging is usually used, under conditions of increased environmental factors (such as humidity, temperature, radiation dose rate [21,22,23,24], due to the impossibility of conducting such tests over many years before the material in question is used. Many reasonable attempts have been made to develop empirical and mathematical models for correlating polymer test results under accelerated aging conditions to aging results under actual service conditions [22,25,26,27]. However, these correlations pose difficulties since in a complex environmental system, where factors such as temperature, oxygen diffusion rate and, finally, the time of evolution of parallel and subsequent chemical processes play a role, the effect of the aging process varies depending on the intensity of the aging factor. Therefore, it is important to determine to what extent the processes occurring on a microscopic scale and underlying the effects observed as material aging are analogous for aging under test conditions to those occurring during authentic operation. This is because the similarity of these processes at the molecular level in both aging conditions guarantees the true comparability of the aging processes and thus the successful application of the aging model.

A simplified diagram of the formation of radicals in PE-type polymer is presented in Figure 1. 

The three types of carbon-centered free radicals, namely, alkyl radicals (−CH_2_−•CH−CH_2_−), allyl radicals (−•CH−CH=CH−), and polyenyl radicals (−CH_2_−•CH−(CH=CH)n−CH_2_−) have been previously reported in the literature [28,29,30]. The alkyl radicals can convert to allyl radicals in the absence of oxygen [31], and upon air exposure, peroxy radicals are immediately formed [32,33]. 

Bolland and Gee’s basic autoxidation scheme (BAS) has long been accepted as the general autoxidation scheme for all polymers. Central to this scheme is the conjecture that damage propagation to the next polymer chain occurs through the cleavage of a hydrogen atom via a peroxy radical. However, this reaction is strongly thermodynamically unfavourable for all polymers except unsaturated ones, where the product allylic radical is resonantly stabilised [10,34]. In the case of saturated plastics such as polypropylene (PP) [35], polyethylene (PE) [36,37], and other polymers [38,39,40], degradation increases with increasing concentration of unsaturated groups. The oxidative degradation of polyethylene, although not as strong as in the case of unsaturated polymers, is nevertheless a well-documented process, which is encountered, among others, in the present study. An explanation was proposed in the paper by Zhao et al. [28], postulating that BAS is valid for saturated polymers, but only at unsaturated defect sites where H transfer is thermodynamically favourable. Another possibility is that peroxy termination is a dominant effect over the H-transfer [41,42]. This dominance of peroxy termination is confirmed in view of the study by Hettal et al. [19] parallel to the one presented here. This research, conducted on the same set of samples (unprotected XLPE, TeamCables aging protocol), reports an increase and subsequent decrease in the concentration of POOH groups in the samples along with aging. This phenomenon is explained by the predominant influence of POOH recombination processes in the samples as POOH concentrations increase. It is noteworthy that this effect coincides perfectly with the effect of an increase and then decrease in the relative concentration of POO• radicals with aging (both regimes) of the unprotected samples tested herein. The described phenomenon indicates the mechanistic, radical-related aspects of the aging of XLPE and is reflected in the macroscopic properties of the studied samples. As shown in Figure 7, for the low dose rate the concentration of POO• radicals for non-protected samples reaches a relative peak against alkyl radicals for the second aging point, after which it decreases. For the high-dose aging, the maximum is reached at 4th aging period. This course correlates with the course of the EaB changes published by Hettal et al. [19] (samples not protected with AO). The EaB values directly correspond to the radical process changes observed here. The maximum POO• vs. Alk• concentration for the low-dose aging, is reached already at the 2nd aging period (followed by the decrease in POO• vs. Alk• and accordingly, the critical EaB value is exceeded for the 2nd low-dose aging period as well. Similarly, with a linear increase of the POO• vs. Alk• concentration in high-dose aging, there is a linear decrease in EaB of the sample. It is notheworthy that the critical EaB 50% value is reached for the penultimate sample displaying maximum POO• vs. Alk• concentration, and exceeded for the last (most aged) sample. 

Although this is only a preliminary observation, the data presented suggest the existence of a relationship between the content and kinetics of radicals in polymer samples, as observed by EPR cryogenic technique, and the physical condition of the polymer, which is the resultant of the chemical processes taking place. This renders the herein-presented analysis an interesting way of predicting polymer parameters and, consequently, of assessing the condition of cables in critical electrical installations. The advantage of the method presented here is the small sample size (a couple of mg), which allows low-destructive condition assessment studies of polymeric products, enabling testing even in areas where additional representative samples are not provided.

The relative concentration of peroxy radicals vs. alkyl radicals in the aged samples is completely different for the different aging dose rates. For aging in high-dose-rate regime (400 Gy/h), one can observe a stable increase in the propensity of POO• vs. Alk• formation, up to 65% of the initial (POO• + Alk•/Alk•) value. Meanwhile, for a dose of 8.5 Gy/h there is a much more pronounced increase in the level of POO• versus Alk• to the second aging step (51 kGy) followed by its gradual decrease in longer aging periods (up to a dose of 120 kGy). The effect of AO content on the level of POO• vs. Alk• radical formation is clearly observable. This effect is retained during aging at both the low and the high dose rates (Figure 7). For aging in the high-dose regime after reaching a dose of 250 kGy, one can observe a depletion of the protective effect of AO. 

It is well-documented that macroradicals formed in amorphous areas react rapidly with each other or with oxygen dissolved in the polymer, while those formed in crystalline areas show high stability [43,44]. We observe a similar effect of the stabilisation of radicals in the studied materials with aging, which may be due to the increase in the degree of crystallinity observed for the samples studied [19]. At the same time, it should be noted that in another study, no significant increase in the level of gel fraction, a parameter considered to be a direct indicator of the degree of cross-linking, was observed for the samples tested in this analysis; however, cross-linking observed at the 995–860 cm^−1^ IR spectral range showed that more crosslinks are formed at the low dose rate [45]. 

The reliability of polymeric materials and predictions of their performance often depend on quantifying oxidation behavior as a function of not only the aging factor dose and rate, but also of sample depth. A clear understanding of oxygen permeation into the polymer is required to model site-dependent oxidation or to predict the contribution of diffusion-limited oxidation conditions in accelerated aging studies [21]. Oxygen diffusivity and solubility are key basic parameters in the description of oxidation and degradation phenomena of polymers [17,21,46]. The samples investigated in this study were aged at ambient-close temperatures (21–47 °C) that differed only slightly on the scale of phenomena occurring in XLPE (glass transition around −100 °C, melting point in the range of 150–170 °C). Our present analysis did not take into account the different depths inside the insulation material (the samples were 0.5 mm thick). Nevertheless, it should be taken into account that oxygen diffusivity in PE can be increased with the increase of temperature by 26 °C to almost by an order of magnitude higher [47]. It must therefore be assumed that the increased temperature for the sample aged at a low dose rate has additionally contributed to the accumulation of oxygen-centered radicals in the sample, which explains the higher POO• vs. Alk• rates for the low-rate-aged material. 

The fate of the paramagnetic products formed after oxidation can be followed in a series of spectra measured at high microwave power (10 mW), when the signals of carbon-centered radicals are partially saturated. Both irradiated samples contain peroxy radical, which decays to 310 K and 280 K for samples aged at 110 °C and 130 °C, respectively. After thermal annealing of the samples, another oxygen-centered radical appeared, namely alkoxy radical –HCO•– being a secondary product of alkyl radical chain reactions. Its g⊥ value is considerably lower than that attributed to peroxy radical, 2.0210 and 2.033, respectively. The signal appears at around 280 K in the spectra registered at 10 mW, in contrast to anisotropic peroxy radical, which was observed immediately upon irradiation.

As seen in Figure 1, Figure 2, Figure 4, Figure 5 and Figure 6, the signal from the buildup of the POO• radical was most pronounced at 130 K. Its subsequent decrease with the increasing temperature can be interpreted as a recombination of the reactive peroxy radical along with the increase of the mobility of particles. The change in the POO• vs. Alk• intensity ratio at 160K, where the POO• radical participation is the highest, in the samples during the aging processes is shown in Figure 7. In both types of aging, a pronounced increase in the peroxy radical intensity versus that of alkyl radical can be seen for the 1–4 aging periods, while only a slight such increase can be seen for the samples protected with AO. 

In unaged samples as well as in the early stages of aging, the ratio of peroxy radicals to alkyl radicals is the lowest and increases with successive stages to decline again for the longest periods of aging. This again implies the sites for POX formation are produced during aging and then gradually consumed to form terminal non-paramagnetic oxidation products. Nevertheless it has to be underlined that the effects of radiation aging highly depend on the combination of the dose rate, the oxygen availability and the temperature of aging. It is also noteworthy that the macroscopic (mechanical) properties of the material correlate with the observed paramagnetic product levels.

## 5. Conclusions

Analysis of the relative intensity of the (POO• + Alk•) to Alk• signal by measuring the intensity at g = 2.0318 and g = 1.9692, respectively, for successive radiation aging stages in the high- and low-dose regimes for XLPE unprotected and protected with AO can provide a measure of oxidative damage occurring with aging and allows us to determine the protective effect of AO. Antioxidants prevented the buildup of the radical oxygen species in the intermediate intervals of aging (Figure 2), probably, by competing with the potential sites for oxygenation in the polymer. The XLPE unprotected with AO was shown to be strongly degraded by gamma irradiation. The dose rate affected the amount and type of macroradicals formed. These differences can be linked to the different oxygen availability in the material, which is a function of temperature and determines availability at a given dose rate. This implies that to obtain a dose-response equivalent to the real aging process, a longer exposure time is required at a low dose rate, which enhances the possibility of oxygen diffusion into the material and, therefore, the degradation of the material. The mechanical (EaB) properties of the XLPE as measured in [19] directly correspond to the POO• vs. Alk• concentrations for both regimes of radiation-induced aging studied herein. The analysis of the peroxy-to-alkyl radical ratio indicated the extent of the protective effect of antioxidants. The addition of 1phr Irganox 1076, and 1phr Irganox PS 802 resulted in POO• radical scavenging in the studied XLPE sheets observed up to the dose of 200 kGy. 

These results are in line with those of other studies performed within the TeamCables project including spectroscopic, dielectric, thermal, and mechanical testing [6,19,45,48,49,50]. Future work on this topic will include the broadening of the scope of the studied aging factors and the direct relation of the obtained results to those from other methods to evaluate cable degradation for broader investigations.

## Data Availability

The data supporting reported results can be found in Appendix A.

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
