# Peer review of "Radical Composition and Radical Reaction Kinetics in the Probe-Irradiated XLPE Samples as a Potential Source of Information on Their Aging Degree"

_materials, 2022, doi:10.3390/ma15165723_

Round 1
Reviewer 1 Report
The authors present a detailed work on the use of EPR to study over time the degree of oxidation and radical fragmentation of polyethylene used to make cable sheaths for nuclear power plants. The objective is of interest because these types of installations continue to operate and are likely to continue to do so for much longer, so developing a relatively simple method that requires a small amount of sample to analyse the durability and suitability of the cables is useful and novel.
In addition, the authors include antioxidant-doped cables to improve the study of the results obtained by this technique.
However, I have some comments for the authors:
- In the introduction, nuclear energy is referred to as a sustainable energy, an issue that has caused controversy when the European Commission has recently considered it as such in the face of the energy crisis in which we are immersed. It is undoubtedly a cleaner source of energy than others, although it is also true that the management of the waste it generates is complicated. I would therefore recommend the authors to soften the sentence in lines 49-51 in this respect.
- I think that the introduction, which is quite long (171 lines) contains few references (only 5), so they should include more; there are paragraphs (such as the first one) in which it is necessary to include references, even if they are of a general nature.
- Throughout the manuscript, the terms "aging" and "ageing" are used indistinctly. In order to maintain the homogeneity of the text, the authors should choose and use only one. The same is true, although to a lesser extent, for "unaged" and "non-aged".
- Figures 3A, 3B, 7A and 7B are cut off on the right, also affecting the legend. They could be made a little smaller to fit in the manuscript.
- On lines 406 and 407 the chemical formula numbers should be put as subscripts.
- In references 5 and 30 the name of the journal is abbreviated, when in the rest of the references it is not. In references 23, 25, 28 and 34 some capital letters are missing in the titles of the journals, and in reference 31 the title of the book chapter appears in capitals, but the name of the book and the publisher or journal in which it was published are missing, as well as the year of publication.
Author Response
We are glad that the reviewer found our work sound. We thank the reviewer for careful and thoughtful reading and revision, which helped us much to improve the manuscript. Please, find below the answers to the issues raised by the reviewer.
- In the introduction, nuclear energy is referred to as a sustainable energy, an issue that has caused controversy when the European Commission has recently considered it as such in the face of the energy crisis in which we are immersed. It is undoubtedly a cleaner source of energy than others, although it is also true that the management of the waste it generates is complicated. I would therefore recommend the authors to soften the sentence in lines 49-51 in this respect.
We agree this is a controversial statement. As not being of much importance here (regardless of its sustainability the NPPs operation times are being prolonged), this sentence was deleted.
- I think that the introduction, which is quite long (171 lines) contains few references (only 5), so they should include more; there are paragraphs (such as the first one) in which it is necessary to include references, even if they are of a general nature.
The introduction has been carefully revised, slightly shortened and redundant sentences removed for clarity. Many new references have been added.
- Throughout the manuscript, the terms "aging" and "ageing" are used indistinctly. In order to maintain the homogeneity of the text, the authors should choose and use only one. The same is true, although to a lesser extent, for "unaged" and "non-aged".
Thank you for careful reading. Indeed, we changed it to ‘aging’ and ‘unaged’ for homogeneity.
- Figures 3A, 3B, 7A and 7B are cut off on the right, also affecting the legend. They could be made a little smaller to fit in the manuscript.
Indeed it is very inconvenient for the reviewer to read, we apologize. Please, find the new version with adjusted figures.
- On lines 406 and 407 the chemical formula numbers should be put as subscripts.
Done, thank you.
- In references 5 and 30 the name of the journal is abbreviated, when in the rest of the references it is not.
The abbreviations were adequately changed to full journal names.
-In references 23, 25, 28 and 34 some capital letters are missing in the titles of the journals, and in reference 31 the title of the book chapter appears in capitals, but the name of the book and the publisher or journal in which it was published are missing, as well as the year of publication.
This has been improved, thank you.
Reviewer 2 Report
The article describes the potential application of the EPR spectroscopy to determine the aging degree of widely used polymers such as XLPE. The authors show how the ratio of signal intensities of the two types of radicals mostly observed (peroxide and alkyl) depends on the aging conditions of the samples and the presence of antioxidants. Undoubtedly, the article focuses on a topic currently of enormous interest and the article will be interesting for the general reader of this publication.
The work has been well planned and executed being experiments and results described in detail. However, the discussion about signal intensity changes with increasing temperature exclusively in terms of recombination and/or transformation into other radicals, without taking into account effects such as increased mobility, reduced saturation effects or decreasing magnetic susceptibility, can be risky. For example, it is common for peroxide radicals in oxides to stop being visible in EPR at RT due to their greater mobility, but they become visible again when the temperature drops. In particular, to confirm some of their conclusions it is essential that the authors show that the observed signals do not change if the temperature is lowered again.
Further comments
“Cryogenic EPR” is not a new or special technique. Conducting EPR studies at low temperatures (even at 4.2 K) is very common.
I suggest use EasySpin to analyze the spectra
Title: XPE or XLPE?
Line 175: “XPLE”
Fig. 3A: “P=0.001 mW”
Line 258: “0.1W”
Line 335: “above 260K”
Author Response
We are glad that the reviewer found our work sound. We thank the reviewer for careful and thoughtful reading and revision, which helped us much to improve the manuscript. Please, find below the answers to the issues raised by the reviewer.
The article describes the potential application of the EPR spectroscopy to determine the aging degree of widely used polymers such as XLPE. The authors show how the ratio of signal intensities of the two types of radicals mostly observed (peroxide and alkyl) depends on the aging conditions of the samples and the presence of antioxidants. Undoubtedly, the article focuses on a topic currently of enormous interest and the article will be interesting for the general reader of this publication.
The work has been well planned and executed being experiments and results described in detail. However, the discussion about signal intensity changes with increasing temperature exclusively in terms of recombination and/or transformation into other radicals, without taking into account effects such as increased mobility, reduced saturation effects or decreasing magnetic susceptibility, can be risky. For example, it is common for peroxide radicals in oxides to stop being visible in EPR at RT due to their greater mobility, but they become visible again when the temperature drops. In particular, to confirm some of their conclusions it is essential that the authors show that the observed signals do not change if the temperature is lowered again.
We thank the reviewer for their insightful analysis and inspiring comments on the EPR method.
Regarding the physical effects in the sample: We had applied the method of annealing and then again freezing the sample before, e.g. in the following work of 2012:
https://www.sciencedirect.com/science/article/pii/S0022286012011611?via%3Dihub
These times we would measure in Dewar inset to the spectrometer cavity until our collaborator, Jarosław introduced us to the EPR cryostatted method using a cryostat that fits in the cavity. We observed that the results did not vary, when the sample was cryostatted directly in the cavity and measured at the given temperature, from those obtained by annealing in the external cryostat and measured in Dewar in liquid nitrogen. As our interests generally are centered around similar radicals in polymers, since then we decided the herein applied method was our method of choice. During annealing, we reach temperatures of 310 K and above, until we generally no longer see any paramagnetic centers. For the analysis, we chose a range of microwave powers for which there is no saturation effect for both signals.
Nevertheless, the reviewer's remark indicates the possibility of making interesting observations, and in our next work we will look for differences between the annealed spectra of samples at higher and lower temperatures at different microwave powers. If it were possible to observe the effects mentioned by the reviewer, even small ones, it would give some really interesting information about the structures of paramagnetic centers and their magnetic properties, and could result in another interesting work.
Further comments
“Cryogenic EPR” is not a new or special technique. Conducting EPR studies at low temperatures (even at 4.2 K) is very common.
Of course we agree. The word ‘new’ in abstract was deleted. We meant the method was, to our knowledge, first time used in this context, but it is certainly not a new method.
I suggest use EasySpin to analyze the spectra
We agree EasySpin is an excellent tool for analyzing spectra but, unfortunately, we have not had the opportunity to use this program until now because we did not have access to matlab (paid access). However, the processing used here was basic and limited to subtracting two spectra from each other. For the next project, we will plan the purchase of this platform.
Title: XPE or XLPE?
Line 175: “XPLE”
Fig. 3A: “P=0.001 mW”
Line 258: “0.1W”
Line 335: “above 260K”
Thank you very much for careful reading. All these mistakes were corrected.
Reviewer 3 Report
The manuscript (materials-1859781) studied the accelerated aging behaviors of XPE using irradiation. These results would be important to understand long time aging behaviors of XPE and the effects of anti-oxidants.
1. The meanings of the digital numbers in Figure 1 and 2 are not clear for me.
2. Figure 3 and Figure 6: Some texts are missing.
3. Figure 7: There are 2 AOX curves.
4. Beside radical types and concentrations, could authors provide extra information of the aged samples, such as mechanical tests and transparency measurements?
Author Response
We are grateful for the reviewer’s remarks, which helped to improve our work. Please, find below the answers to the issues raised by the reviewer.
- The meanings of the digital numbers in Figure 1 and 2 are not clear for me.
Thank you, indeed the graphs may be not readable, these numbers denote measurement temperatures. The graphs have been corrected in this regard (unit K has been added) and the comment in the caption has been made.
- Figure 3 and Figure 6: Some texts are missing.
The units in figures were added, the explanations on g-values were added
- Figure 7: There are 2 AOX curves.
The two AOXLPE are each for one of the two dose rates studied. The graph A is in the function of aging time, B, in the function of dose.
- Beside radical types and concentrations, could authors provide extra information of the aged samples, such as mechanical tests and transparency measurements?
The additional data regarding inter alia mechanical properties and surface spectral data of the same samples were measured by other groups of the consortium. They can be found in references [6,19,45,48–50]. Generally we can say the changes in these properties follow similar patterns. Particularly, a discussion on the occurrence of POOH in regard to POO· concentration and on the EPR signal stability in view of crystallinity changes in these samples is included [referred to Hettal, 6, 19]. A comment on this has been made in conclusions.